# Male applicants are more likely to be awarded fellowships than female applicants: A case study of a Japanese national funding agency

**Daisuke Kyogoku**[1]*, **Yoko Wada**[2]

**1** The Museum of Nature and Human Activities, Sanda, Hyogo, Japan, **2** Faculty of Agriculture, Miyazaki University, Miyazaki, Japan

* d.kyogoku@gmail.com

**Data Availability Statement:** The data underlying the analyses is included in the submission as supplementary information.

## Abstract

Scientific grant applications are subjected to scholarly peer review. Studies show that the success rates of grant applications are often higher for male than for female applicants, suggesting that gender bias is common in peer review. However, these findings mostly come from studies in Europe, North America and Australia. Here we report the analyses of gender-specific success rates of applications to the fellowships offered by Japan Society for the Promotion of Science (JSPS). Because we analyze the observational data (i.e., not experimental), our aim here is to describe the possible gender gaps in the success rates, rather than the examination of gender bias per se. Results show that the success rates are consistently higher for male applicants than for female applicants among five different fellowship categories. The gender gaps in the success rates varied significantly between research fields in some Fellowship categories. Furthermore, in some fellowship categories, the gender gaps were significantly associated with the representation of female applicants (both positive and negative correlations were found). Though the causes of the gender gaps are unknown, unintentional gender bias during the review process is suggested. Pre-application gender gaps may also be contributing to the gender gaps in success rates. At least some of the observed gender gaps were relatively small, which may be partly explicable by the designs of the review process. However, gender gaps or biases acting prior to the application, such as self-selection bias, may have reduced the superficial gender gaps in the success rates. Further investigations that control for the effects of covariates (e.g., scientific merits of each applicant, which were not accessible to us) and those of other funding agencies, especially of non-Western countries, are warranted.

## Introduction

Scholarly peer review is the cornerstone of modern science. Peer review is generally thought to act as the gatekeeper, assuring the quality of published literature. Peer review is also involved in the evaluation of grant applications. The quality of peer review is thus crucial to the development of science. However, the quantitative understanding of the peer-review process is

**Funding:** The authors received no specific funding for this work.

**Competing interests:** The authors declare no conflict of interests.

immature [1], and there could be potential shortcomings in peer review. For example, the review process can be gender-biased [2]. Because publication records and grants awarded are important measures of success for scientists, the potential gender bias can put some people at a disadvantage for reasons irrelevant to science.

The pioneering work of Wennerås and Wold [2] showed a substantial gender gap in the success rates in a Swedish research fellowship, directing the interest of scientists to the roles of gender biases in peer-review processes. Wennerås and Wold [2] controlled for the effects of important covariates (i.e., scientific productivity), and a substantial gender gap remained after that, suggesting a gender bias. Many subsequent studies have analyzed gender gaps in the success rates of research grants. The meta-analysis of these studies found a robust gender gap in the success rates, with female applicants being less successful [3]. However, most studies examined the grant applications to European, North American or Australian funding agencies. This is an important knowledge gap, because gender gaps and/or gender discrimination in science is found in other countries or regions too, including South Africa [4], Brazil [5], and Southeast Asia [6].

Japan is one of the unique countries in terms of gender in science. Japan has a relatively long and successful history in modern science among non-Western countries (e.g., [7]). Meanwhile, the substantial gender gaps in science in Japan have been pointed out many times (e.g., [8–10]). For example, the Organization for Economic Cooperation and Development (OECD) reported that Japan had the lowest share of female staff in tertiary education in 2020 (30%) among 32 comparable member countries of the OECD [11] (see also [12]). More generally, the representation of women in the Japanese labor market is low, despite legislative efforts. In Japan, the Basic Law for a Gender-Equal Society was enacted in 1999, which provides general guidelines for promoting gender equality in society. Furthermore, several laws were enacted primarily to improve the working environment for women (e.g. Equal Employment Opportunity Act, Act on Childcare Leave·Caregiver Leave, Act on Promotion of Women's Participation and Advancement in the Workplace in 1985, 1991, and 2015, respectively). However, the female labor participation rate is still well below the average of the OECD countries [13]. This cultural background suggests that gender gaps in grant peer review are likely in Japan. Here we report the analyses of gender-specific success rates of applications to the fellowships offered by a Japanese national funding agency.

Japan Society for the Promotion of Science (JSPS) offers several categories of fellowships to young scientists of all scientific disciplines (including social sciences and humanities). JSPS Research Fellows are considered academic elites in Japan, and thus JSPS Fellowships are taken as paths toward successful academic careers. Here we analyze the data of five Fellowship categories. Overseas Fellowship is a two-year fellowship for PhD holders planning to join an overseas research group. PD Fellowship is a three-year fellowship for PhD holders based at a Japanese institution. Both Overseas and PD are for young PhD holders, and those who finished their PhD in the last five years are eligible (parental leave excluded). RPD Fellowship is a three-year fellowship for those who are back to research after parental leave (no constraint on academic age). These three postdoctoral Fellowships can be applied for by those with Japanese nationality or the right of permanent residence. DC1 Fellowship is a three-year fellowship for first-year PhD students. DC2 Fellowship is a two-year fellowship for second-year or more senior PhD students. DC1 and DC2 can be applied for by students based in a Japanese institution, regardless of the applicants' nationality. There are (were) other fellowship categories, but only these five categories have been there in the last five years; we thus limit the analyses to these five categories (the details of JSPS Fellowships in the past are digitally archived by the Web Archiving Project of National Diet Library, Japan: https://warp.ndl.go.jp/, retrieved on April 9, 2023).

Applications to JSPS Fellowships are reviewed by six reviewers. JSPS defines nine Large-Scale Research Fields (e.g., engineering, biology). Large-Scale Fields are subdivided into Middle-Scale Fields, which are further subdivided into Fine-Scale Fields. Applicants choose the Fine-Scale Fields to apply, and professional scientists from the Fine-Scale Fields are chosen as the reviewers. Each reviewer independently scores applications. The applications that earned high scores in this first round of screening are appointed the Fellowship. The applications with intermediate scores are subjected to the second round of screening (the designs of the second screening have not been the same in the last five years). Application forms include personal information (name, affiliation, etc), a research plan, an achievement summary, and a recommendation letter(s). Gender-specific numbers of overall applications and appointed applications are publicly available on the website of JSPS, enabling the analysis of gender-specific success rates.

In this study, we analyze the gender gaps in the success rates of JSPS Fellowship applications. Though we are interested in gender bias during the peer-review process, this study is not experimental (i.e., observational). The gender gap in success rates can emerge without gender bias if male and female applicants show systematic differences in their scientific merits or any other relevant factors. The main question we address here is thus whether there are gender gaps in the success rates. We first examine the overall gender gap for each fellowship category, without considering the differences between research fields. We then examine how gender-specific success rates vary between research fields (at Large-Scale classification). In order to better understand the patterns of gender-specific success rates, we also examine the correlation between gender gaps in the success rates and the representation of females among applicants, because the unconscious bias against females is known to be strong when females are a minority among those who are evaluated [14]. We discuss possible causes underlying the patterns found by the analyses.

## Data and methods

### Dataset

We used the data publicized on the website of JSPS. On the website, the numbers of overall applications and appointed applications are reported, separately for male and female applicants. This enables the calculation of gender-specific success rates. We compiled the data for the last five years on March 2, 2022 (URL: https://www.jsps.go.jp/j-pd/pd_saiyo.html). There are several categories of JSPS Fellowship, as has been explained in the Introduction. On the website, the numbers of applications and successful ones are also reported separately for each research field (at Large-Scale classification), which were also compiled.

### Methods

We first examined whether there were gender gaps in application success rates. We constructed generalized linear mixed models (GLMMs) with a binomial error structure with a logit link function. The response variable was the application success rate (i.e., the GLMMs explain the probability parameter of the binomial distribution, with "numbers of trials" corresponding to the gender-specific number of applicants), and the fixed effect was the applicant's gender. A random effect was the calendar years; we assumed the year-by-year fluctuation in success rates as a random noise without a consistent trend or autocorrelation. The significance of the gender difference was examined using a likelihood ratio test (LRT) by dropping the gender term. The analysis, as well as all the following analyses, was performed separately for the five Fellowship categories.

We also examined whether the possible gender gaps in the success rates varied among the research fields. Research field categorization was renewed in 2018 (i.e., this policy has been applied to the Fellowship appointed from 2019), as a routine renewal. This makes the comparison before and after this reform difficult. We thus used the data from the last three years for this analysis. We constructed GLMMs whose response variable was the success rate. In this analysis, the fixed effects included the applicant's gender, research field, and their interaction. Years were included as a random effect. The significance of the research field-specific gender gaps was examined using the LRT by dropping the interaction term.

Finally, we examined the factors that were associated with the research field-specific gender gaps (if any) in the success rates. Specifically, we examined whether year- and research field-specific gender gaps in the success rates were associated with the absolute numbers of female applicants or the proportions of female applicants. For this analysis, we first obtained year- and research field-specific gender-gap effect sizes (with their estimation errors) by making general linear models (GLMs) with a binomial error structure. The response variable was the success rate, and the explanatory variable was applicants' gender. We applied this model to subsets of data for each year and each research field. We compiled the estimated effects of gender and their Wald standard errors of the mean (SEM) from the estimated GLMs. The estimates were then used as the response variable of a GLMM with a normal distribution (the estimates were weighted by the inverse of SEM). The data points were given weights so that unreliable outliers do not skew the regressions. The fixed effect was the absolute number of female applicants or the proportion of female applicants, and the random effects were research fields and years. The significance of the fixed effects was examined using LRT by dropping the fixed effect term.

## Results

### Overall gender gaps

In all categories of JSPS Fellowships, male applicants were more successful than female applicants (Fig 1A–1E). These gender gaps were all statistically significant but RPD (Overseas, $\chi^2$ = 10.836, $d.f.$ = 1, $P < 0.001$; RPD, $\chi^2$ = 1.096, $d.f.$ = 1, $P$ = 0.295; PD, $\chi^2$ = 10.673, $d.f.$ = 1, $P$ = 0.001; DC2, $\chi^2$ = 21.047, $d.f.$ = 1, $P < 0.001$; DC1, $\chi^2$ = 7.690, $d.f.$ = 1, $P$ = 0.006, Fig 1, S1–S5 Tables in S1 File). There were few male applicants to RPD, which resulted in large uncertainty in the gender gap estimate (S2 Table in S1 File). The estimates of the gender gap tended to increase with the career stage; whereas the gender gaps were relatively small among DC1 and DC2 applicants (i.e., students), those among PD, RPD and Overseas applicants (postgraduate researchers) were large (Fig 1F). Overall proportions of female applicants varied among Fellowship categories (five-year average and minimum and maximum in the five years were: Overseas 20.1% [18.4–22.1%], RPD 94.4% [93.6–95.3%], PD 27.7% [26.9–29.2%], DC2 26.2% [25.4–26.6%], and DC1 24.4% [23.6–25.9%]).

### Research field-specific gender gaps

Gender gaps in the success rates varied significantly between research fields among the Overseas and PD applicants (Overseas, $\chi^2$ = 24.214, $d.f.$ = 8, $P$ = 0.002; PD, $\chi^2$ = 17.324, $d.f.$ = 8, $P$ = 0.0269). Among the PD applicants, throughout the three years, female success rates were consistently lower than those of male applicants in the research fields of "Mathematical and Physical Sciences", "Chemistry", "Engineering Sciences" and "Medicine, Dentistry and Pharmacology" (Fig 2A). Among the Overseas applicants, throughout the three years, female success rates were consistently lower than those of male applicants in the research fields of "Mathematical and Physical Sciences" and "Medicine, Dentistry and Pharmacology" (Fig 2B).

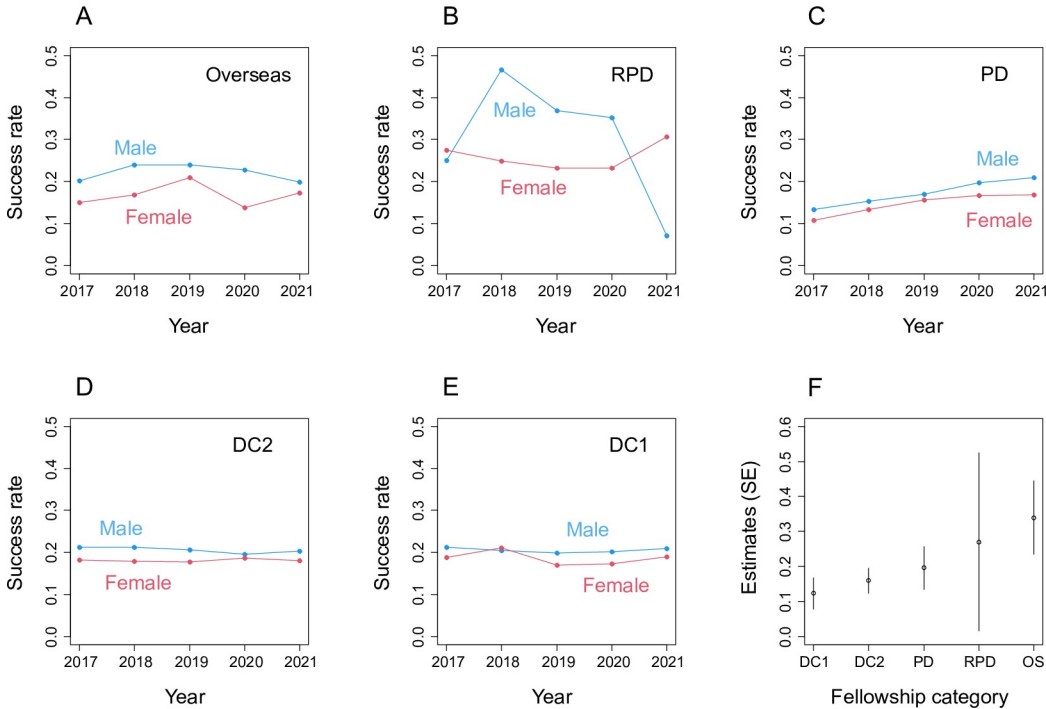

**Fig 1. Gender gaps in the success rates of JSPS fellowships.** (A)–(E) Success rates of male and female applicants to different Fellowship categories. (F) GLMM estimates (log odds ratio) of the gender gaps in the success rates for different Fellowship categories (SE is the Wald standard error). Positive estimates represent higher success rates of male applicants than female applicants.

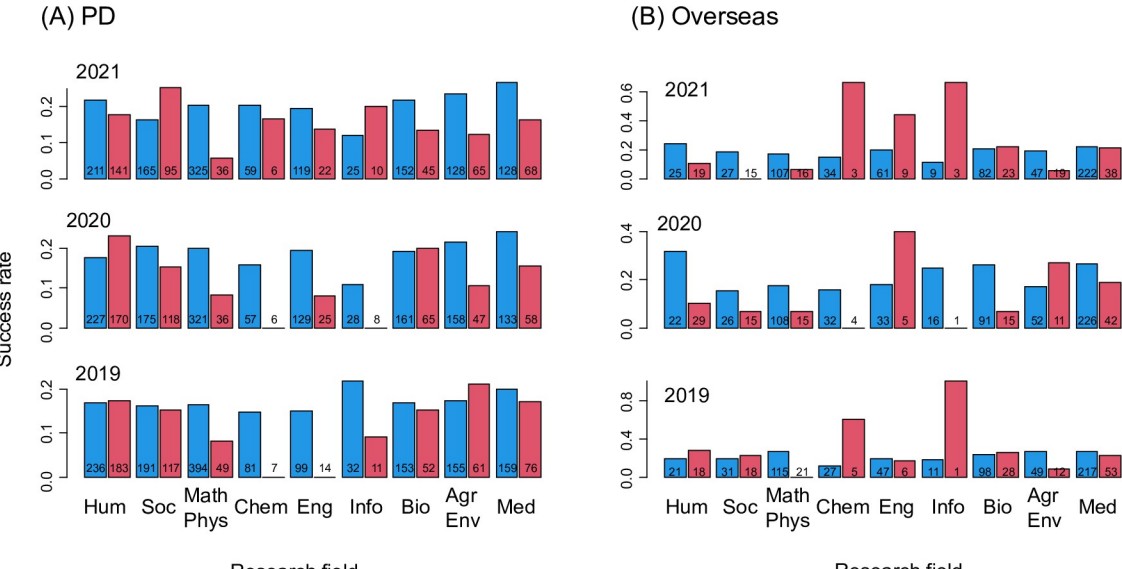

**Fig 2.** Gender- and research field-specific success rates of PD (A) and Overseas (B) applicants. Male (left) and female (right) success rates are shown in a pair for each research field (in JSPS Large-Scale classification). The numbers on the bars represent the number of applicants. Hum: Humanity, Soc: Social Sciences, Math Phys: Mathematical and Physical Sciences, Chem: Chemistry, Eng: Engineering Sciences, Info: Informatics, Bio: Biological Sciences, Agr Env: Agriculture and Environmental Science, Med: Medicine, Dentistry and Pharmacology.

In no research field, female applications were consistently more successful than male applications. On the other hand, gender gaps in the success rates of DC1 and DC2 applicants did not differ significantly between research fields (DC1, $\chi^2 = 3.678$, $d.f. = 8$, $P = 0.885$; DC2, $\chi^2 = 9.424$, $d.f. = 8$, $P = 0.308$; S1 Fig in S1 File). RPD was excluded from this analysis, because few male applicants (no male application in several research fields) made the GLMM parameter estimation difficult (S1 Fig in S1 File).

### Gender gaps and female applications

In the analysis of PD, where the response variable was the year- and research field-specific gender gap in the success rate, the result showed that female applicants were less successful when there were fewer female applicants ($\chi^2 = 5.001$, $d.f. = 1$, $P = 0.0253$, Fig 3A, S6 Table in S1 File). When the number of female applicants was replaced with the proportion of female applicants in the GLMM, the estimated standard deviation for the random effect of year was very small ($< 0.001$) and the parameter estimation failed. When this random effect was dropped, female applicants to PD were less successful when the proportion of female applicants was smaller ($\chi^2 = 11.041$, $d.f. = 1$, $P < 0.001$, Fig 3B, S7 Table in S1 File). The similarity of these results is not surprising, because the number of female applicants and the proportion of them showed a strong positive correlation (Fig 3C). One may reckon that this correlation is trivial (i.e., more female applicants automatically increase their proportion among all applicants). However, this is not necessarily the case. Male application numbers and their proportions showed only an ambiguous correlation (Fig 3D). Indeed, the number of male applicants did not show a significant correlation with the gender gaps ($\chi^2 = 0.482$, $d.f. = 1$, $P = 0.487$, Fig 3E, S8 Table in S1 File).

In similar analyses for Overseas applicants, female success rates did not show significant correlations with the number of female applicants ($\chi^2 = 2.300$, $d.f. = 1$, $P = 0.129$) or the proportions of them ($\chi^2 = 1.301$, $d.f. = 1$, $P = 0.254$) (see also S9–S11 Tables in S1 File). The results for DC2 applications were qualitatively similar (female application number: $\chi^2 = 0.163$, $d.f. = 1$, $P = 0.687$; proportions of female applicants: $\chi^2 = 1.043$, $d.f. = 1$, $P = 0.307$, S12–S14 Tables in S1 File).

In a similar analysis of DC1 applicants with the number of female applications, estimated standard deviations of the random effects of the research field and year were $< 0.00001$, and GLMM parameter estimation failed. The analysis without the random effects (i.e., GLM) showed that the gender gap in the success rates was higher when there were more female applicants ($\chi^2 = 5.633$, $d.f. = 1$, $P = 0.018$, Fig 4A, S15 Table in S1 File). The analysis with the proportions of female applications, which included the two random effects, similarly showed that the gender gap increased with increasing proportions of female applicants ($\chi^2 = 15.05$, $d.f. = 1$, $P < 0.001$, Fig 4B, S16 Table in S1 File). The number of female applicants and the proportion of them showed a strong positive correlation (Fig 4C), whereas male application numbers and their proportions showed a less clear correlation (Fig 4D). Parameter estimation for the GLMM with the number of male applicants failed (random effect standard deviation for year was $< 0.001$), and we dropped the random effect of years. The resulting GLMM showed that the gender gap decreased with increasing male application number ($\chi^2 = 4.137$, $d.f. = 1$, $P = 0.042$, Fig 4E, S17 Table in S1 File).

### Discussion

There were consistent gender gaps in the success rates of JSPS Fellowships; female applicants were less likely to be awarded the Fellowships than male applicants (Fig 1). The overall gender gap among RPD applicants was not statistically significant, but the estimated gender gap was

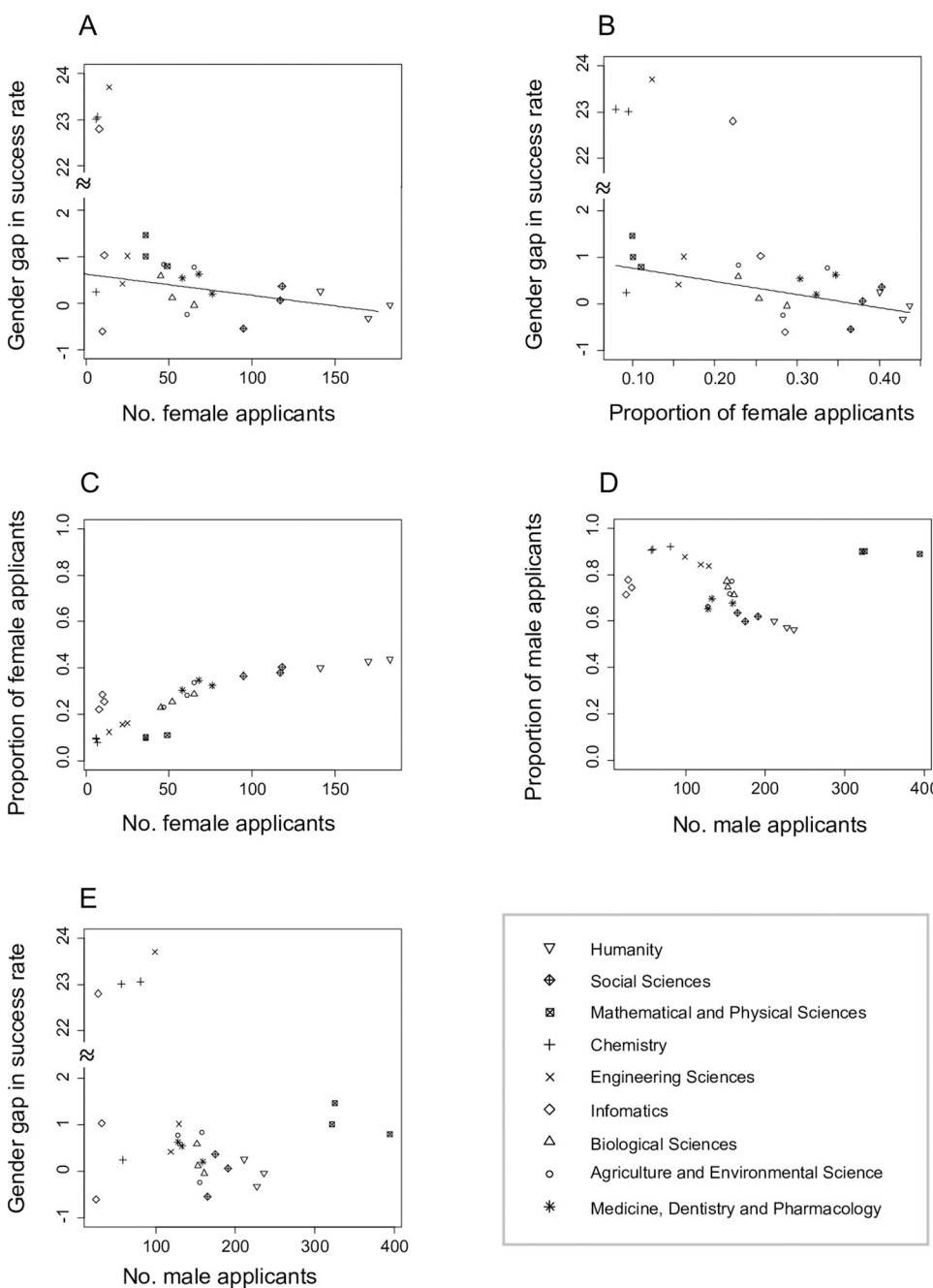

**Fig 3. The gender gap in success rates of PD application and female application sizes.** (A) Year- and research field-specific gender gaps against the absolute numbers of female applications. (B) Year- and research field-specific gender gaps against the proportions of female applications. (C) The correlation between female application numbers and their proportions. (D) The correlation between male application numbers and their proportions. (E) Year- and research field-specific gender gaps against the absolute numbers of male applications. Gender gaps are estimates of GLMs (i.e., log odds ratio; see Data and methods for details); positive values indicate that male applications are more successful than female ones. Outliers in A, B and E with very large gender gaps are associated with large standard errors, and thus given little weight in the regression analyses.

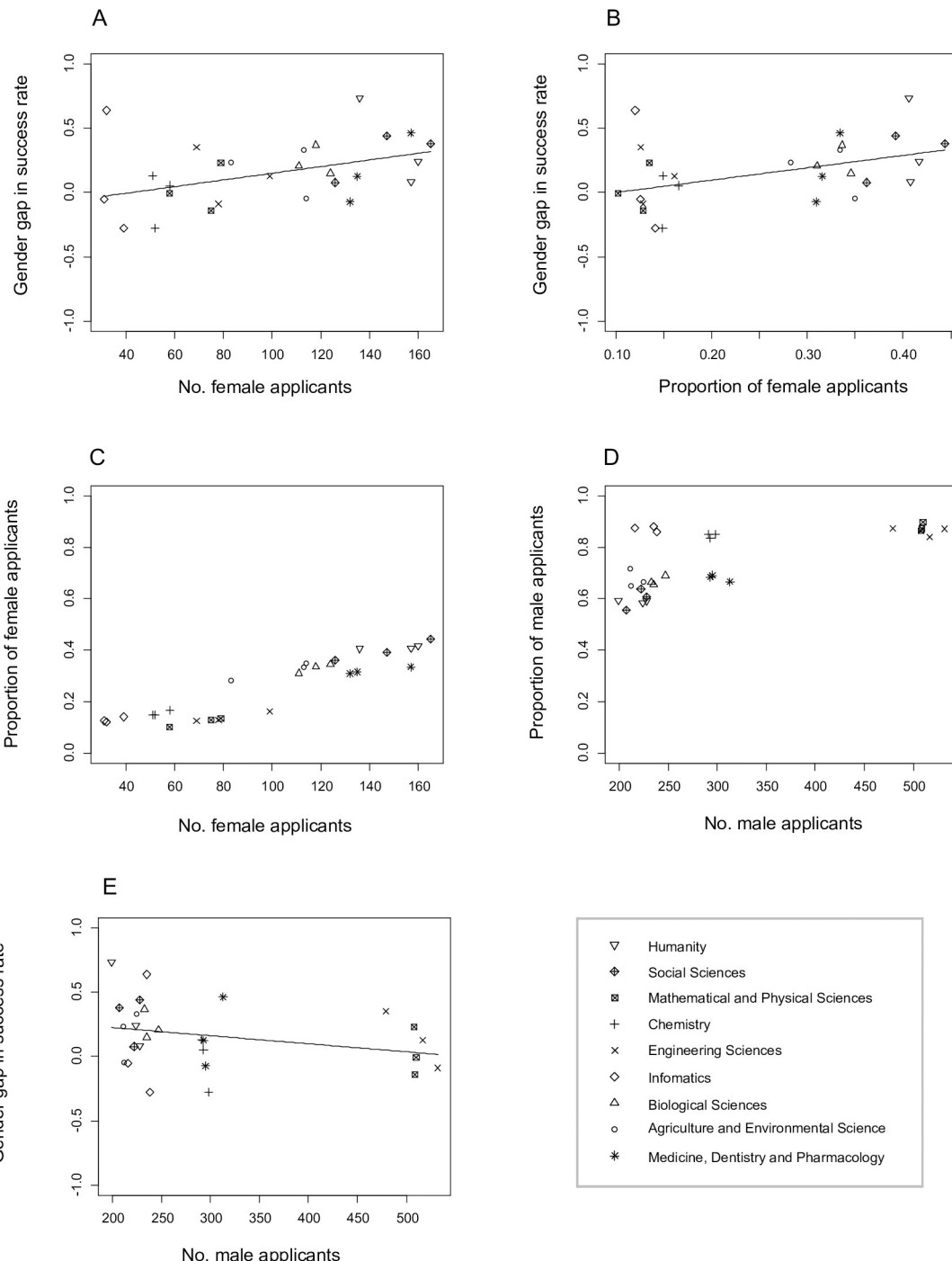

**Fig 4. The gender gap in success rates of DC1 application and female application sizes.** (A) Year- and research field-specific gender gaps against the absolute numbers of female applications. (B) Year- and research field-specific gender gaps against the proportions of female applications. (C) The correlation between female application numbers and their proportions. (D) The correlation between male application numbers and their proportions. (E) Year- and research field-specific gender gaps against the absolute numbers of male applications. Gender gaps are estimates of GLMs (i.e., log odds ratio; see Data and methods for details); positive values indicate that male applications are more successful than female ones.

comparable to those of other Fellowship categories. Few male applications to RPD (S1 Fig in S1 File) made the uncertainty in the gender gap estimate large (Fig 1F). Though the gender gaps were relatively small, they tended to be larger in the Fellowship categories for later career stage researchers (PD, RPD and Overseas) than for earlier career stage researchers (DC1 and DC2). Gender gaps in the success rates of PD and Overseas applicants varied significantly among research fields (Fig 2). Among PD applicants, the gender gaps were large when there were fewer female applications (Fig 3). A similar pattern was not found among Overseas applicants. Surprisingly, more female applications to DC1 were significantly associated with larger gender gaps (Fig 4). Below we discuss hypotheses that may explain these patterns. We also discuss the implications and limitations of our findings and future research directions.

A factor likely contributing to the overall gender gaps is gender bias during the Fellowship review process. We do not believe that the reviewers are intentionally discriminating against female applicants. Yet, many studies have shown that evaluation processes are prone to unintentional, implicit and unconscious biases. Experimental studies have repeatedly demonstrated that applicants' gender, inferred by their names, influences the evaluation outcome (e.g., [15–18]). Postulated gender bias in the review process appears consistent with the reduced gender gaps with increasing female applications seen among PD applicants. Gender bias against females is known to be exacerbated when the proportion of females is small among the evaluated people [14]. Though this hypothesis cannot explain why the gender gaps among Overseas applicants do not decrease with female applications, other processes such as self-selection bias may be confounding (see below). Overall gender gaps tended to be larger for postgraduate Fellowships than for graduate Fellowships (Fig 1F), and this appears also consistent with the postulated gender bias during the review process. Witteman et al. [19] found that gender bias during the review processes of scientific grants is larger when the reviewers focus on the scientific competence of the applicants, as opposed to the proposed science. At least from 2017 through 2021, the review for PD (i.e., postgraduate fellowship) for example put emphasis on achievements, whereas that for DC1 and DC2 (graduate fellowship) put emphasis on future prospects (though the details of Application Guidelines have not been the same during this period; past Application Guidelines are archived online [Web Archiving Project, National Diet Library, Japan]). One may assume achievements (e.g., number or quality of publications) are objective measures. However, studies show that female achievements tend to be attributed to good luck, whereas male achievements tend to be attributed to their abilities [14]. Therefore, the review process of PD with emphasis on achievements might exacerbate the postulated gender bias.

Another possible explanation for the overall gender gaps in the success rates is that male applicants are more competent than female applicants on average. Evidence indicates that such a complex trait as competence as a scientist is unlikely to show a systematic gender difference for physiological reasons [20–22], though there are some gender differences in very specific skills seemingly due to physiological reasons (e.g., mentally rotating 3-dimensional objects; Valian [14]). However, social factors might make male applicants more competent (or competitive) than female applicants. For example, PhD advisors may assign better scientific projects to male students than to female students, or male students may accumulate publications more quickly than female students. Recent studies showed that female scientists' contributions are less acknowledged than male scientists' (e.g., not receiving authorship [23, 24]), which can result in a publication record discrepancy between genders [25, 26]. In addition, typical traits of males such as self-promotion and overt competitiveness may be considered favorable by the reviewers [27]. If this is the case, measures of competence that do not favor a particular gender may be recommended [26–28]. Alternatively, male applications may be supported by better recommendation letters, as is repeatedly shown [29–32]. Yet,

recommendation letters play less important roles in Japanese academia than in Europe or North America [33], making the potential role of recommendation letters obscure.

Despite the consistency, the observed gender gaps in the success rates were relatively small at least for some Fellowship categories. The effect sizes of gender (log odds ratio) were 0.12 and 0.16 for DC1 and DC2, respectively (S4 and S5 Tables in S1 File). The effect sizes compiled in the meta-analysis by Bornmann et al. [3] ranged between −0.2 and 0.25, and their overall effect size across studies was 0.07 (positive values favoring males). Though log odds ratio does not uniquely map to a particular value of percentage point difference, these results suggest that the observed gender gaps in our study are comparable to the previously reported ones and not exceptionally large. The relatively small gender gaps may be explicable by the design of the peer-review procedure. During the JSPS Fellowship review process, tens of applications are reviewed by a group of six experts (each member independently scores the applicants). Reviewers score each application on a relative scale, as opposed to an absolute scale. A recent study shows that gender bias is mitigated when multiple applicants are compared simultaneously than when they are evaluated separately [34]. In addition, reviewers of JSPS Fellowships are required to describe the reasons for their review scores. Accountability is known to reduce bias during evaluation processes [35]. These designs of JSPS Fellowship review process may be working to reduce possible gender bias. However, we also note that the effect size for Overseas (0.40) was unprecedentedly large (though, again, the interpretation of the log odds ratio is not straightforward). As noted in the Introduction, few studies have examined the gender gaps in the grant peer-review of non-Western countries. Future investigations of grant peer reviews in diverse countries would not only better contextualize the current results but also provide a broader view of gender gap distribution in the world.

Relatively small gender gaps found in this study could be an underestimation of the possible gender bias during the review process; other factors may be reducing the superficial gender gaps. For example, self-selection bias may discourage applications by females (i.e., only highly competent females may be willing to apply). In 2017, 33% of PhD students were female in Japan (Gender Equality Bureau Cabinet Office; https://onl.tw/tmKZSgP, May 5, 2022 retrieved). The proportions of female applicants were consistently lower than this among all the JSPS Fellowship categories but RPD, suggesting that males are more likely to apply for JSPS Fellowships than females. This is not surprising, as males are known to be more confident and risk-prone than females independently of their competence [36–40]. Notably, the proportions of female applicants were lower among DC1 and DC2 applicants (graduate students) than among PD applicants (postgraduate researchers). This implies the confounding effect of self-selection bias on the observed gender gaps in success rates; the postulated self-selection bias in the application to DC1 and DC2 implies that female applicants may be on average more competent than male applicants. This should increase the superficial success rates of female applicants. Indeed, the gender gaps were relatively small in DC1 and DC2. Larger gender gaps in DC1 success rates with more female applications appear also consistent with this idea. Small self-selection bias will increase female applications, which will equalize the competence of male and female applicants (as opposed to higher competence of female applicants), exposing similarly competent male and female applicants to the postulated gender bias during the review process (but it should be noted that the variation in female applicants' representation in Fig 4 should also reflect the number/proportion of female students in each field). The same argument (the effect of self-selection bias) appears also applicable to Overseas applicants (there were relatively few female applicants), but the difference between graduate and postgraduate applications will complicate the interpretation.

Here we reported the consistent gender gaps found among JSPS Fellowship applicants. Though this result suggests gender bias during the peer review, the causes of the gender gaps

are unknown, because the effects of important covariates were not controlled. For example, an analysis that controls for the scientific performance of the applicants (e.g., number and quality of publications) would reveal the relative contributions of the gender gaps that pre-exist prior to the application and gender bias. Applicants' racial and ethnic backgrounds may also influence the review outcomes [41, 42]. In addition, it should be noted that the extent of gender gaps can change over time or cultural background [14, 43–45]; the gender gaps in JSPS Fellowships may have been larger before and become smaller in the future. These possibilities warrant further investigations. In any case, describing this kind of gender gap is the crucial first step to better science culture, especially in a country like Japan, where the gender gaps are substantial as was described in the Introduction. The postulated gender bias during the review process calls for remedies, and remedy implementation should be guided by evidence [46]. For example, one intuitive remedy may be to increase the proportion of female reviewers, but many studies have repeatedly shown that females also show gender bias against females (e.g., [16, 18, 42]). We also note the importance of openness. This study was possible because the data were publicly accessible. We suggest keeping the gender-specific data publicly accessible to make future investigations possible.

## Supporting information

**S1 Data.**
(XLSX)

**S1 File.**
(PDF)

## Acknowledgments

Ryosuke Ito, Yuko Kurita, Yasuyuki Nomura, Yumeki Oto, Satsuki Tsuji, Ryo Yamaguchi and JSPS kindly provided information about the Application Guidelines of the Fellowships. YW is grateful to Itsuro Koizumi for the literature information.

## Author Contributions

**Conceptualization:** Daisuke Kyogoku.

**Data curation:** Daisuke Kyogoku.

**Formal analysis:** Daisuke Kyogoku.

**Investigation:** Daisuke Kyogoku.

**Methodology:** Daisuke Kyogoku.

**Project administration:** Daisuke Kyogoku.

**Visualization:** Daisuke Kyogoku.

**Writing – original draft:** Daisuke Kyogoku.

**Writing – review & editing:** Daisuke Kyogoku, Yoko Wada.

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
