## [Decision Letter · Decision Letter 0]

4 Apr 2023

PONE-D-22-28352Male applicants are more successful than female applicants at the fellowships of a Japanese national funding agencyPLOS ONE

Dear Dr. Kyogoku,

Thank you for submitting your manuscript to PLOS ONE. After careful consideration, we feel that it has merit but does not fully meet PLOS ONE’s publication criteria as it currently stands. Therefore, we invite you to submit a revised version of the manuscript that addresses the points raised during the review process.

We look forward to receiving your revised manuscript.

Kind regards,

Claudia Noemi González Brambila, Ph.D.

Academic Editor

PLOS ONE

Journal Requirements:

"No. The funders had no role in study design, data collection and analysis, decision to publish, or preparation of the manuscript."

"There is no conflict of interest to declare."

Reviewers' comments:

Reviewer's Responses to Questions

**Comments to the Author**

1. Is the manuscript technically sound, and do the data support the conclusions?

Reviewer #1: Partly

Reviewer #2: Yes

2. Has the statistical analysis been performed appropriately and rigorously? 

Reviewer #1: Yes

Reviewer #2: I Don't Know

3. Have the authors made all data underlying the findings in their manuscript fully available?

Reviewer #1: Yes

Reviewer #2: Yes

4. Is the manuscript presented in an intelligible fashion and written in standard English?

Reviewer #1: Yes

Reviewer #2: Yes

5. Review Comments to the Author

Reviewer #1: General Comments:

1. under the abstract section the purpose of the research and the research methodology should be stated explicitly. Besides, research implication/recommendation should be included.

2. there are some paragraphs with out citation. So, they should be cited

3.under methodology section the specific methods used should be justified scientifically

4. There is a mistake also in the referencing

5. the title also needs to be modified

6. replace 'I' with 'Researcher'

Reviewer #2: It is important to discuss cultural aspects based on the historical evolution of the regions. Gender discrimination is observed in studies on the role of women in science in Africa, Latin America and Southeast Asia. Another important aspect to consider is the change in gender roles and the incorporation of women into the labor market. This would enrich the discussion of the article. Another important aspect is the consideration of the nationality of the applicants, which can also play an important role in the decision to accept or not a proposal. If these aspects are not considered, the statistical discussion could lose impact. Another element to consider is the comparative element with other offices or institutions that grant similar financing and from other regions.

6. PLOS authors have the option to publish the peer review history of their article (what does this mean?). If published, this will include your full peer review and any attached files.

Reviewer #1: **Yes: **

Reviewer #2: No

---

## [Author Response · Author response to Decision Letter 0]

31 May 2023

Please see our response letter file.

---

## [Decision Letter · Decision Letter 1]

29 Aug 2023

Male applicants are more likely to be awarded fellowships than female applicants: a case study of a Japanese national funding agency

PONE-D-22-28352R1

Dear Dr. Kyogoku,

We’re pleased to inform you that your manuscript has been judged scientifically suitable for publication and will be formally accepted for publication once it meets all outstanding technical requirements.

Kind regards,

Claudia Noemi González Brambila, Ph.D.

Academic Editor

PLOS ONE

Additional Editor Comments (optional):

Reviewers' comments:

Reviewer's Responses to Questions

**Comments to the Author**

1. If the authors have adequately addressed your comments raised in a previous round of review and you feel that this manuscript is now acceptable for publication, you may indicate that here to bypass the “Comments to the Author” section, enter your conflict of interest statement in the “Confidential to Editor” section, and submit your "Accept" recommendation.

Reviewer #1: (No Response)

Reviewer #2: All comments have been addressed

2. Is the manuscript technically sound, and do the data support the conclusions?

Reviewer #1: Yes

Reviewer #2: Yes

3. Has the statistical analysis been performed appropriately and rigorously? 

Reviewer #1: Yes

Reviewer #2: Yes

4. Have the authors made all data underlying the findings in their manuscript fully available?

Reviewer #1: Yes

Reviewer #2: Yes

5. Is the manuscript presented in an intelligible fashion and written in standard English?

Reviewer #1: Yes

Reviewer #2: Yes

6. Review Comments to the Author

Reviewer #2: The author successfully responded to the comments and observations. The article improved substantially and is ready to be published

7. PLOS authors have the option to publish the peer review history of their article (what does this mean?). If published, this will include your full peer review and any attached files.

Reviewer #1: 

Reviewer #2: No

---

## [Editor Report · Acceptance letter]

26 Sep 2023

PONE-D-22-28352R1 

Male applicants are more likely to be awarded fellowships than female applicants: a case study of a Japanese national funding agency 

Dear Dr. Kyogoku:

I'm pleased to inform you that your manuscript has been deemed suitable for publication in PLOS ONE. Congratulations! Your manuscript is now with our production department. 

Kind regards, 

on behalf of

Dr. Claudia Noemi González Brambila 

Academic Editor

PLOS ONE